# Neutron reflectometry and NMR spectroscopy of full-length Bcl-2 protein reveal its membrane localization and conformation

Ameeq Ul Mushtaq [1,6], Jörgen Ådén [1,6], Luke A. Clifton [2], Hanna Wacklin-Knecht[3,4], Mario Campana[2], Artur P. G. Dingeldein[1], Cecilia Persson[5], Tobias Sparrman[1] & Gerhard Gröbner [1✉]

B-cell lymphoma 2 (Bcl-2) proteins are the main regulators of mitochondrial apoptosis. Anti-apoptotic Bcl-2 proteins possess a hydrophobic tail-anchor enabling them to translocate to their target membrane and to shift into an active conformation where they inhibit pro-apoptotic Bcl-2 proteins to ensure cell survival. To address the unknown molecular basis of their cell-protecting functionality, we used intact human Bcl-2 protein natively residing at the mitochondrial outer membrane and applied neutron reflectometry and NMR spectroscopy. Here we show that the active full-length protein is entirely buried into its target membrane except for the regulatory flexible loop domain (FLD), which stretches into the aqueous exterior. The membrane location of Bcl-2 and its conformational state seems to be important for its cell-protecting activity, often infamously upregulated in cancers. Most likely, this situation enables the Bcl-2 protein to sequester pro-apoptotic Bcl-2 proteins at the membrane level while sensing cytosolic regulative signals via its FLD region.

[1] Department of Chemistry, University of Umeå, Umeå, Sweden. [2] ISIS Pulsed Neutron and Muon Source, Science and Technology Facilities Council, Rutherford Appleton Laboratory, Harwell Science&Innovation Campus, Didcot, Oxfordshire, UK. [3] European Spallation Source ERIC, ESS, Lund, Sweden. [4] Department of Chemistry, Division of Physical Chemistry, Lund University, Lund, Sweden. [5] The Swedish NMR Center, University of Gothenburg, Gothenburg, Sweden. [6] These authors contributed equally: Ameeq Ul Mushtaq, Jörgen Ådén. ✉email: gerhard.grobner@chem.umu.se

Programmed cell death, also called apoptosis, is essential for embryonic development, tissue homeostasis and removal of harmful cells in mammals and many other multi-cellular organisms[1,2]. Following activation of the intrinsic apoptotic pathway by intracellular stress, mitochondria play major roles in the process through permeabilization of the mitochondrial outer membrane (MOM) system and consequent release of apoptotic proteins such as cytochrome c from their intermembrane spaces[3]. This finally paves the irreversible way towards cellular self-destruction[4]. A key adaptive mechanism that protects healthy cells from inappropriate death, but eliminates harmful cells, is tight regulation (positive and negative) of this apoptotic pathway and MOM integrity by members of the Bcl-2 (B-cell lymphoma 2) protein family[3,5,6]. Pro- and anti-apoptotic proteins of this family meet at the MOM and their net interactions determine the host cell's fate: maintenance of intact MOM and cell survival or MOM perforation and cell death. The outcome is governed by their relative abundances and affinities to each other[7–9], which are strongly affected by the local membrane environment[10]. Disturbances of this regulatory interplay can cause various pathological disorders, including cancer[11,12].

The main Bcl-2 members are multidomain proteins and are generally thought to regulate commitment to apoptosis by controlling the MOM's integrity[13]. Therefore, these proteins must be membrane-active. They can be classified as 'tail-anchored membrane proteins'[14,15], since they contain beside their main globular fold composed of amphitropic domains, also a single transmembrane domain (TMD) at their C-terminus, which is thought to facilitate their binding and localization to their target membranes. The post-translational mechanism by which these proteins translocate upon ribosomal generation to their specific intracellular membranes is not fully understood yet[15,16]. Nevertheless, various Bcl-2 proteins can become soluble by hiding their TMD inside their hydrophobic BH3 binding groove motif. Subsequent exposure of the TMD enables them to insert into their target membrane and adapt an active membrane-embedded conformational state[15,17]. This way apoptotic members such as Bax are kept soluble in the cytosol and only migrate to, and permeabilize the MOM following activation and exposure of their TMD, leading to partial membrane penetration[9,18]. Even the anti-

apoptotic Bcl-x$_L$ is found in various soluble and membrane-anchored subpopulations in cells[16,19]. In contrast, despite the structural similarity, the anti-apoptotic, name-giving Bcl-2 protein itself is only found firmly anchored to intracellular organellar membranes, most prominent to the MOM system[14,16]. Intact human Bcl-2 was found to be very insoluble (presumably due to a fully exposed TMD), and therefore its subcellular translocation to the MOM is not yet clear. However, it is suggested that this process actively involves other proteins like e.g. mitochondrial chaperones[16,20]. In the membrane Bcl-2 protects healthy cells simply by sequestering apoptotic proteins to prevent perforation of the MOM and release of apoptotic factors such as cytochrome c[10]. Bcl-2 has a specific groove region which is central in restraining these apoptotic proteins (Bax, Bak, and BH3 only proteins) by binding to their Bcl-2 homology 3 (BH3) motifs. This cell-protection mechanism also plays a notorious role in tumorigenesis and cancer treatment resistance, as the Bcl-2 protein is often upregulated by mechanisms that block death signals[12].

Nevertheless, all these anti-apoptotic Bcl-2 proteins should—based on their "tail-anchored" sequence homologies—adopt a globular fold composed of six amphipathic helical domains wrapped around two more hydrophobic central helices and the membrane-anchoring TMD. This combination provides them with the conformational flexibility to transit from a cytosolic-like to a loosely membrane-tethered state and finally to a firmly membrane-integrated conformational state required to exert their function[14,15]. However, the molecular basis of this cell-protection functionality of anti-apoptotic Bcl-2 proteins is poorly understood, due to lack of knowledge of their membrane-associated state and topology[10,21].

Recent structural work on soluble anti-apoptotic Bcl-x$_L$ indicated that membrane-anchorage by its C-terminus alone is not sufficient for its inhibitory functionality[19]. Thus, a membrane-embedded conformation is also presumably required for those multidomain anti-apoptotic Bcl-2 proteins to function[15]. However, we lack molecular understanding of this mechanism due to the challenges and inherent complexities associated with these conformationally highly flexible proteins in their cellular membrane environment.

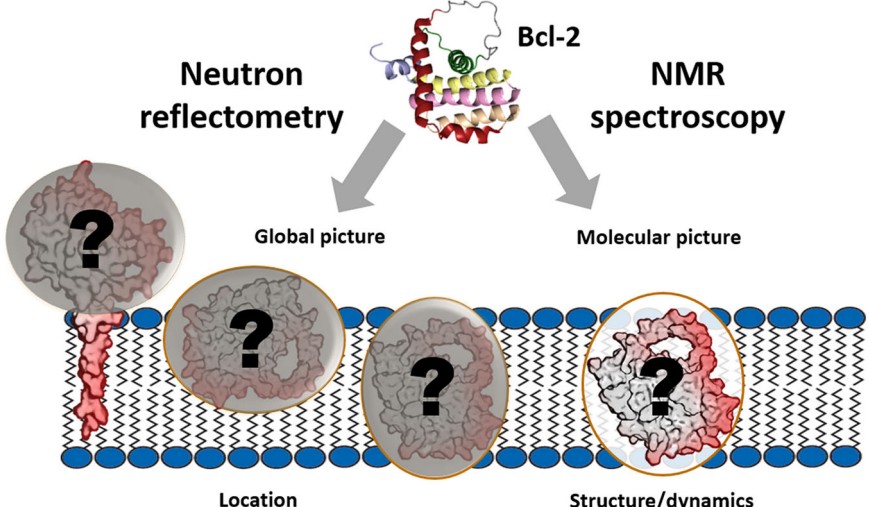

**Fig. 1 Schematic diagram of approaches used to probe the Bcl-2 protein's structure and dynamics in a membrane environment.** NR (neutron reflectometry) provides a global picture of the membrane, including locations of its molecular compounds and their volumetric contributions, as shown in this schematic diagram of the Bcl-2 protein (in tail-anchored and membrane-embedded positions). NMR provides complementary structural and dynamic information at atomic to entire protein molecule levels. The schematic Bcl-2 structure shown is based on a model of Bcl-2 (derived from PDB_ID 1YSW), including its fully flexible loop domain (FLD).

To address the lack of molecular understanding of the membrane-associated functionality of anti-apoptotic Bcl-2 proteins we have studied both the location and the structural and dynamic characteristics of the full-length human Bcl-2 protein in a mammalian membrane-mimicking environment. For this, we used two complementary techniques: neutron reflectometry (NR) and NMR spectroscopy, as schematically illustrated in Fig. 1. NR has become a powerful technique for probing biological membranes of varying complexity in recent years since it can provide abundant information on the structure and composition of entire membrane systems and their molecular components[22,23].

*Inter alia*, it can provide indications of proteins' distributions across a lipid membrane and thus their locations in it. This type of global information is not accessible by other methods such as x-ray crystallography and NMR spectroscopy. However, NMR can provide—at atomic resolution—structural and dynamic descriptions of the behavior of specific molecular (and even intramolecular) components, including lipids and proteins[24,25]. Solid-state NMR approaches are especially suitable for probing unmodified, full-length proteins in their membrane-complexes at near-physiological conditions[26–28].

Since full-length (239 aa) human Bcl-2 protein is insoluble and difficult to produce, soluble truncated and chimeric (166 aa) forms have been used in most studies[29–31]. As these forms lack key functional features, like the membrane-anchoring C-terminus and regulatory FLD, we developed a protocol to produce mg amounts of fully functional, intact human Bcl-2 protein[32]. NR analysis of this protein reported here, revealed the location and compactness of the intact Bcl-2 protein in its membrane environment, with most of the protein embedded in the membrane like an integral membrane protein. Complementary solid- and liquid-state NMR experiments corroborated this depiction. NMR-derived data also indicate that the protein's regulatory (ca. 58 aa) loop domain is highly flexible, consistent with a location at or near the membrane interface, while the main protein body is deeply buried and motionally restricted in the membrane.

## Results

### Neutron reflectometric evidence of the Bcl-2 protein's membrane location

For NR analyses, full-length human Bcl-2 protein was reconstituted in supported L-α-dimyristoylphosphatidylcholine (DMPC) lipid bilayers at a 70:1 lipid-to-protein molar ratio. The Quartz Crystal Microbalance with Dissipation (QCM-D) technique was used to develop a protocol for membrane deposition and protein incorporation with NR reflectometry data (Fig. 2a) confirming that well-aligned supported DMPC/Bcl-2 and pure DMPC bilayers could be generated.

The protein and lipids in the bilayers could be resolved by isotopic labeling of the individual membrane components (protonated/fully deuterated Bcl-2 and DMPC, and vice versa) due to the large difference in neutron scattering length density of hydrogen and deuterium. To obtain a full range of membrane depth profiles (Fig. 2 and Supplementary Figs. 1, 2), we acquired a series of co-refined NR datasets for d-Bcl-2/h-DMPC and h-Bcl-2/d-DMPC in media with various $H_2O/D_2O$ buffer ratios (Table 1 and Supplementary Table 1). Representative profiles of deuterated Bcl-2 protein in a DMPC membrane are shown and compared with the profile of a pure DMPC bilayer in Fig. 2a. The profiles indicate that in both systems the membranes have very high quality, and have clearly differing scattering patterns that directly indicate the presence, or absence, of embedded Bcl-2 protein.

The analysis was carried out using Abelès matrix formalism[33], with the proteo-lipid-membrane structure described using a series of homogeneous layers across the bulk interface. The obtained NR signal is modeled as the sum of contributions from each layer and the reflectivity arising from the bulk interface, which forms the SLD profiles across it, as shown in Fig. 2b. The presented SLD profile corresponds to the best model fit of the data and shows their high quality. It clearly indicates that the main part of Bcl-2 resides in the membrane's hydrophobic interior (where its volumetric contribution is ca. 9%; Table 1) and the rest (presumably its outer parts, including the flexible loop region) is located in the headgroup/interface region (volumetric contribution: ca. 7%). These results support the general model, with most of the protein deeply embedded in the host membrane and its more amphipathic/hydrophilic regions located at the interface region. The NR profiles in Fig. 2 do not show any protein density outside the membrane.

### Solid-state NMR: behavior of Bcl-2 in its membrane setting

Solid-state NMR experiments with dynamic filtering[24,34] support the NR-derived model of a compact protein body buried in the host membrane. These involved cross-polarization (CP) NMR experiments to visualize motionally restricted (often membrane-embedded) protein parts and insensitive nucleus enhancement polarization transfer (INEPT) NMR experiments to probe very dynamic and flexible protein segments, as usually found in aqueous or extra-membranous environments. This approach has been recently applied in analyses of GPCR (G-protein coupled receptors) membrane proteins and membrane-associated Bax protein[24,34].

As shown in Fig. 3a, strong, overlapping $^{13}C$ CP MAS NMR signals acquired at 278 K indicate that the vast majority of Bcl-2 residues in the DMPC membranes are dynamically restricted. This confirms the main protein body's location in a tight membrane hydrophobic core, which severely restricts reorientation dynamics of the amino acid residues located in it. Dynamically restricted residues of the Bcl-2 body seem to form a helix bundle across our experimental temperature range region. Substantial changes in the NMR spectra with temperature increases (indicated by sharp lipid NMR signals) only appear at 313 K (Fig. 3a, the spectrum of protein-free DMPC lipid bilayers for comparison). This is due to the melting of the DMPC bilayer matrix above 297 K into its biologically relevant liquid-crystalline phase.

### NMR insights into the location and dynamics of Bcl-2's regulatory FLD region

The Bcl-2 protein contains a nearly 60 aa long intrinsically disordered region: the FLD between its α1 and α2 helices (aa 32-90). This domain is one of the functional hotspots of Bcl-2 and is thought to act as an autoregulatory molecular switch similar to the loop region of its close relative Bcl-$x_L$[35–37]. The FLD sequence of Bcl-2 is more hydrophilic than the main body of the protein and presumably located in the water-accessible membrane surface region[35]. Since this domain represents only a minor, loosely organized part of the protein with a low volume fraction, as manifested by smearing of the layer profile (Fig. 2b), it would be difficult to detect by NR outside the membrane. This difficulty is exacerbated by roughness (heterogeneity in interfacial position) of the planar membrane, being ~4 Å for the d-Bcl-2 (Bcl-2 containing) membrane.

However, if this intrinsically disordered loop region displays high conformational flexibility, its residues can be discerned by NMR with dynamic filtering, as described above. As expected, the mobile loop residues emerge as sharp NMR resonances in the $^{13}C$ INEPT MAS NMR spectra, even at low temperatures (Fig. 3b, 278 K), in contrast to the motionally restricted residues of the protein body visualized by CP NMR (Fig. 3a,c). Upon increasing the temperature to 313 K, these $^{13}C$ INEPT NMR signals become more pronounced and additional sharp protein signals arise due

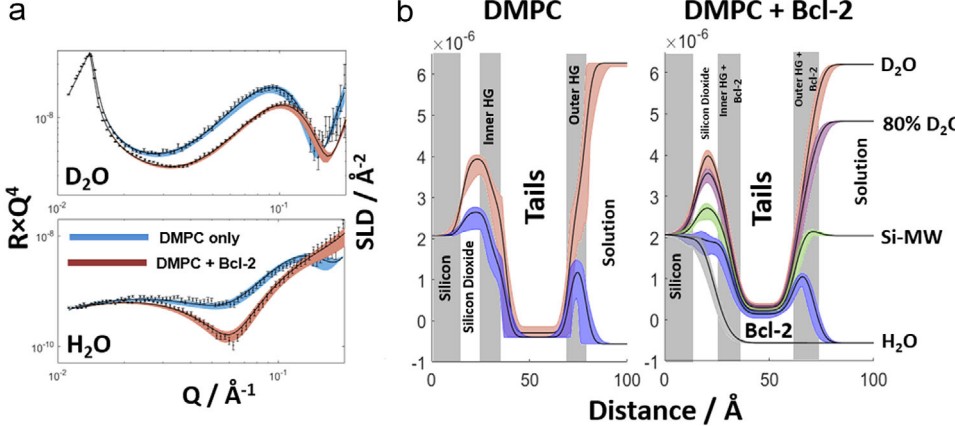

**Fig. 2 Neutron reflectometry (NR) derived membrane profiles in the presence of Bcl-2 protein.** Experimental NR profiles (error bars) and model data fits (lines) obtained for a fully deuterated Bcl-2:fully protonated-DMPC (at a 70:1 lipid-to-protein molar ratio) supported lipid bilayer under multiple solution isotopic contrast conditions. Reflectivity profiles are shown in Reflectivity × $Q^4$ to highlight features in the data arising from the interfacial structure. **a** Comparison of experimental datasets and model-to-data fits for DMPC bilayers with and without Bcl-2, where the SLD (scattering length density) contrast between the solution and the lipid and protein components was highest (in the presence of $D_2O$ and $H_2O$ buffers, respectively). The results highlight the effects of the presence of Bcl-2 in the bilayer on the experimental reflectivity profiles and subsequent model-to-data fits. **b** Scattering length density profiles obtained from all solution isotopic contrast model-to-data fits for the DMPC only (left) and the composite DMPC Bcl-2 membrane (right) datasets. Line widths in both the fits and SLD profiles represent the 95% confidence intervals obtained by MCMC (Markov Chain Monte Carlo) sampling of the model-to-data fits. Overview of results of all NR experiments with varying isotopic contrast conditions and the full set of analyzed data are shown in Supplementary Figs. 1, 2.

**Table 1 Structural parameters for supported DMPC bilayers prior to and in the presence of the Bcl-2 protein.**

| Parameter | DMPC bilayer | DMPC + Bcl-2 bilayer |
|---|---|---|
| Headgroup thickness | 8.3 Å (7.7 Å, 9.5 Å) | 8.2 Å (7.8 Å, 8.8 Å) |
| Headgroup components | DMPC 85% (75%, 88%) -Water 15% (12%, 25%) | DMPC 72% (62%, 80%) Bcl-2 7% (4%, 10%) Water 21% (10%, 34%) |
| Thickness of tails | 32.6 Å (31.8 Å, 33.7 Å) | 30.6 Å (29.7 Å, 31.3 Å) |
| Components of tails | DMPC 98% (96%, 100%) - Water 2% (0%, 4%) | DMPC 90% (88%, 92%) Bcl-2 9% (8%, 10%) Water 1% (0%, 2%) |
| Bilayer roughness | 2.8 Å (1.0 Å, 4.5 Å) | 4.2 Å (4.0 Å, 4.5 Å) |

Values in brackets are confidence intervals as used for fits to the data presented in Fig. 2. If not stated otherwise, 95% confidence intervals were used.

to a further increase in local conformational dynamics. To further visualize these sharp resonances unambiguously in the NMR spectra, and identify the corresponding individual flexible residues in the FLD region, we also carried out solid-state 2D CP $^{13}$C-$^{13}$C DARR and INEPT-based $^{13}$C-$^{1}$H HETCOR NMR experiments (Fig. 3c, d). These mobile signals are not present in the DARR spectrum, but they are highly visible and well-resolved in two dimensions in the HETCOR spectrum. To assign the residues additional solution NMR experiments were performed on U-[$^{13}$C,$^{15}$N,$^{2}$H] labeled Bcl-2 protein in membrane-mimicking dodecylphosphocholine (DPC) micelles. To verify that the protein was fully functional under those conditions we carried out titration studies using a Bax-derived BH3 peptide known to bind to the main groove region of Bcl-2 responsible for recognizing apoptotic proteins via their BH3 motifs[38], as well as the Nur77 peptide known to bind to the flexible loop domain of Bcl-2 protein[35]. As seen in the NMR titration series (Supplementary Figs. 3, 4) the protein is fully functional by binding firmly to those peptides. As expected for the interaction with the Bax-BH3 peptide major chemical shift perturbations (CSP's) could be observed, most pronounced for residues plausibly directly involved in the binding event (s. numbered residues in Supplementary Fig. 3) and less pronounced for most other residues of Bcl-2. This clearly indicates that binding of the peptide to the central part of the Bcl-2 protein, its extended BH3 domain-binding groove, causes major implications across the entire protein fold. In contrast, binding of the Nur77 peptide to the flexible loop domain of Bcl-2 is causing only local changes, as visible in CSP of residues belonging mainly to the loop region while the remaining protein core is much less or not at all affected (Supplementary Fig. 4). Using fully functional Bcl-2 protein in DPC micelles we were finally able to identify nearly 60 amino acids (Supplementary Fig. 5) positioned in the predicted loop region of the 26 kDa protein in its membrane-mimicking micellar environment. Based on these data, various sharp NMR resonances in the HETCOR spectra from membrane-embedded Bcl-2 could be residue-specifically identified (as indicated in Fig. 3d) as parts of the intrinsically disordered loop domain (FLD).

To confirm that these residues are flexible and not dynamically restricted, unlike the main protein body, we conducted $^{15}$N $T_2$ relaxation experiments on full-length $^{13}$C/$^{15}$N uniformly labeled Bcl-2 protein in DPC micelles using a 410 ms CPMG (Carr-Purcell-Meiboom-Gill) pulse sequence[39]. The black spectrum (Fig. 4a) shows signals originating from the rigid protein body and loop region, while the red spectrum only displays signals from the loop region due to its slower relaxation. Those signals are not well dispersed, as expected for residues of IDP-like proteins[40]. Thus, we can unambiguously differentiate between NMR signals originating from residues of the regulatory FLD of the protein and the rigid protein core. The differentiation and conclusions are validated by comparison of these spectra with those of Bcl-2 protein in larger Brij-35 micelles (Fig. 4b).

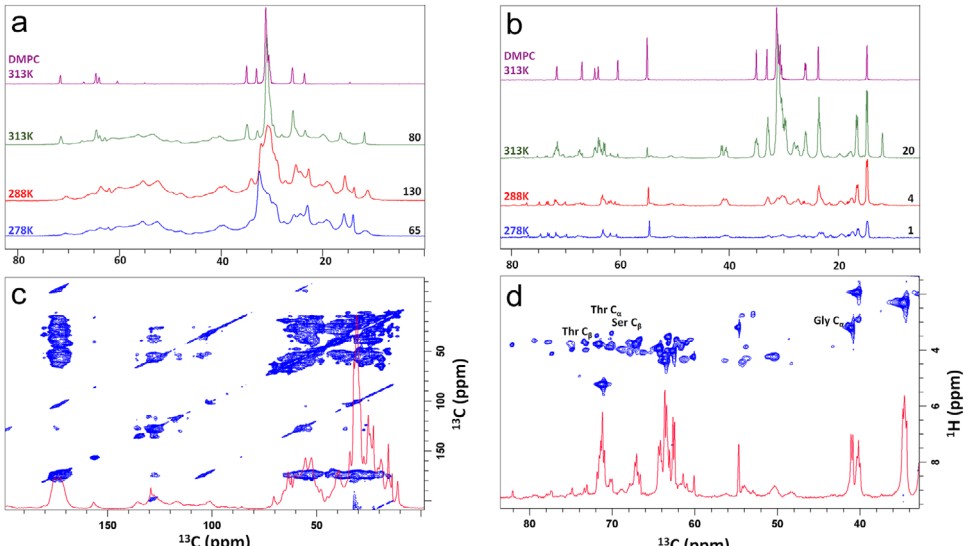

**Fig. 3 Solid-state NMR spectra of Bcl-2 proteoliposomes.** $^{13}C$ CP MAS NMR (**a**) and INEPT MAS NMR spectra (**b**) of U-[$^{13}C$,$^{15}N$] Bcl-2 reconstituted in DMPC membranes at a 30:1 lipid-to-protein molar ratio with indicated temperatures. Expanded spectral regions are shown between 0 to 80 ppm (**a**, **b**), 0 to 200 ppm (**c**) and 35 to 80 ppm (**d**). Rigid (low mobility) fractions of the protein and its membrane environment were probed through CP NMR while flexible regions were detected by INEPT NMR. CP-based 2D $^{13}C$-$^{13}C$ correlation DARR (dipolar assisted rotational resonance) spectrum (**c**) with 100 ms mixing time showing signals from the rigid regions of the protein and (**d**) $^{1}H$-$^{13}C$ INEPT-based 2D HETCOR (heteronuclear correlation) spectrum of a uniformly $^{13}C$,$^{15}N$-labeled sample showing signals from the protein's flexible regions. All NMR spectra were acquired at a MAS spinning rate of 15 kHz and $^{1}H$ frequency of 850 MHz.

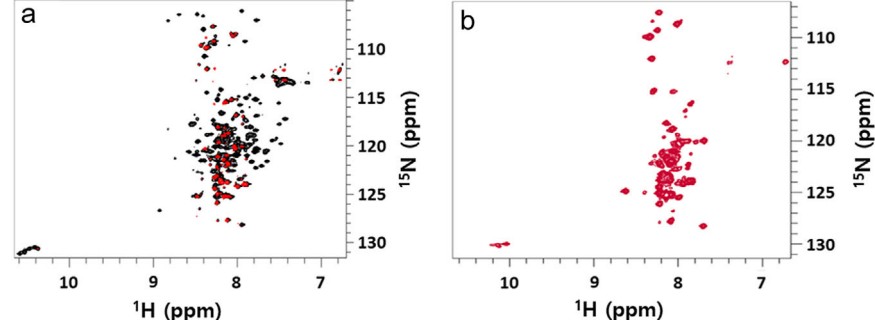

**Fig. 4 Identification of dynamic protein regions.** Relaxation NMR experiments: **a** $^{1}H$-$^{15}N$ TROSY-HSQC spectrum (black) of 300 μM $^{13}C$-$^{15}N$-labeled Bcl-2 in 5 mM DPC micelles at 310 K, with overlaid $^{1}H$-$^{15}N$ TROSY-HSQC spectrum (red) using $^{15}N$ T$_2$ relaxation filtering (410 ms CPMG delay). **b** $^{1}H$-$^{15}N$ TROSY-HSQC spectrum of 270 μM $^{13}C$-$^{15}N$-labeled Bcl-2 in 0.05% (w/v) Brij-35 micelles at 298 K and at a $^{1}H$ frequency of 850 MHz. Signals from flexible loop region residues are visible in red (**a**) and are the only visible signals in (**b**).

In this environment, only the sharp signals from the loop regions with their typical IDP dispersion are visible in $^{1}H$-$^{15}N$ TROSY-HSQC (transverse relaxation-optimized spectroscopy-heteronuclear single quantum coherence) spectra, due to slower relaxation rates, while most signals of the protein core are broadened beyond detection. Signals from the loop residues are also visible as sharp lines in the corresponding $^{13}C$ solid-state MAS NMR spectra of the Bcl-2 proteoliposomes (Fig. 3b, d).

## Discussion

**Implications for Bcl-2 protein functionality in its MOM.** Our combined NR and NMR studies provide detailed molecular insights into the basic structural and dynamic features of Bcl-2 protein. Knowledge of these features is essential to elucidate its cell protection activity at the MOM, where Bcl-2 proteins engage in the regulation of apoptosis. In a physiological lipid membrane environment, full-length Bcl-2 behaves as a membrane-embedded protein (Fig. 5), which is presumably essential for its functions at the mitochondrial membrane, as suggested for anti-apoptotic

Bcl-2 family members[14,15,41]. As seen here by NR, the main protein body (without its loop region) is buried in the membrane, where motions of its residues are restricted, as highlighted by solid-state $^{13}C$ CP NMR experiments (Fig. 3a, b). Solution NMR experiments with Bcl-2 in micelles of varying size confirm the restricted dynamics of these residues (broad lines). In contrast, residues of the very flexible FLD loop region have fast dynamics and high conformational flexibility, manifested by narrow NMR signals (Figs. 3, 4) in both solid-state and solution NMR experiments, due to much longer spin-spin relaxation times caused by fast intramolecular dynamics.

The picture of a fully membrane-embedded Bcl-2 protein body is not surprising since the Bcl-2 protein consists of two central hydrophobic helices (α5, α6) surrounded by a further six hydrophobic/amphiphilic helices, and is linked to a hydrophobic transmembrane anchoring helix region (α9). Since all of these helical structures have affinities for the insides of hydrophobic membrane environments, to various degrees[29,42], a location inside a lipid bilayer with only the loop region sticking out into

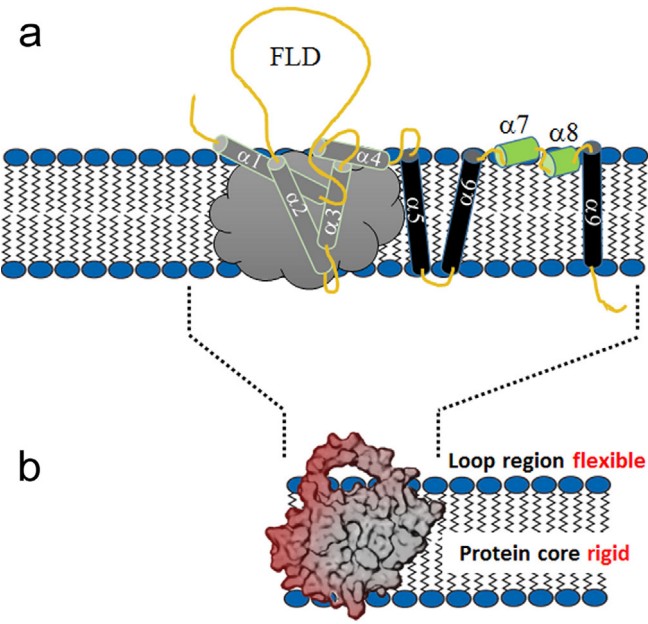

**Fig. 5 Schematic diagram of the membrane-embedded location required for Bcl-2 protein's functionality. a** Arrangement of helices and loops of full-length human Bcl-2 protein derived from its sequence, structural information from soluble Bcl-2 versions[29,31], alterations in membrane topology during apoptosis[14], comparison with the related Bcl-x_L protein (which has conformational plasticity ranging from soluble to embedded states[15,19,41]) and the NR/NMR results reported here. For membrane-inserted α1 to α4 helices (gray cloud) no detailed information is available. **b** Locations of the Bcl-2 main body within the membrane boundaries and solvent-accessible regulatory loop region (FLD) at the membrane surface, observed by NR and NMR, respectively. The main protein body is dynamically restricted while the FLD region is highly flexible.

the aqueous exterior is consistent with expectations. MD simulations comparing truncated versions with our physiologically intact Bcl-2 protein indicate that it is slightly less compact with the large flexible loop region, but still properly folded[36].

Clearly, the function of Bcl-2 as an integral membrane protein observed here must be reconciled with the common view of the nature of anti-apoptotic Bcl-2 proteins' membrane association as deduced from the observation of a soluble member of the family, the Bcl-x_L protein and its behavior upon anchoring to lipid nanodiscs[19]. This involves tail-anchored membrane attachment via their C-terminal hydrophobic transmembrane helix[14,15], while the N-terminal head domain remains in the cytosol by retention of its water-accessible globular fold. Nevertheless, helices 1 and 2 of Bcl-x_L are reportedly close to the membranes in these systems, indicating a propensity for membrane association or insertion. Not surprisingly, previous studies using truncated, soluble Bcl-2 and Bcl-x_L variants were both found to be able to form channels in membranes, most likely due to similar features as pore-forming domains of colicins and diphtheria toxins[43,44]. However, this pore formation seems to be transient without a major effect on the MOM, but presumably prevents permeabilization of those membranes by apoptotic proteins like Bax. While those studies provide no direct information about the Bcl-2 location and exact arrangement in the membrane, they show that Bcl-2 proteins can insert into membranes and form water-accessible pores. Our findings show that in contrast to truncated constructs, the full-length native Bcl-2 protein is embedded firmly into a membrane with its—by helices composed —main fold residing in the membrane interior and only the loop region being water-accessible. Our model (Fig. 5) provides a

fundamental understanding of the unique capability of Bcl-2 protein to recognize apoptotic proteins at the membrane level while accessing via the loop region regulative signals from the cytosol.

Generally, the N-terminal domains of these Bcl-2 members can be water-accessible or membrane-bound. Biophysical studies on various anti-apoptotic Bcl-2 proteins have indicated that they can undergo conformational changes from a cytosolic accessible compact globular fold to a membrane-inserted state[14,41,45]. As summarized by a graphic model (Fig. 1) and accompanying text in a previous publication[15], this conformational plasticity is essential for the proteins' adoption of a membrane-embedded structure, which might be required for its cell-protective functions at the MOM.

Despite intact human Bcl-2 being very insoluble and its translocation mechanism from a cytosolic milieu to the mitochondrial target membrane is not yet clear, our observations of the Bcl-2 protein in an experimental lipid membrane settings are consistent with such conformational plasticity and final membrane-embedment (Fig. 5). By monitoring the behavior of full-length Bcl-2 (239 aa), and a soluble variant (Bcl-2 ΔTM; aa 1-207) with the C-terminal anchor removed, in the presence and absence of DPC detergent we could observe this type of restructuring of the Bcl-2's globular fold into a membrane-like state. As seen in Supplementary Fig. 6, Bcl-2 ΔTM displays a typical solution-like structure with a broad distribution of Trp signals (Supplementary Fig. 6b), as observed for soluble truncated Bcl-2 protein[30], and its Bcl-x_L[19,46] and Bcl-w relatives[45].

When soluble Bcl-2 ΔTM is treated with increasing concentrations of DPC to mimic a membrane environment we observe a structural transition from a soluble to membrane-like embedded state using $^1$H-$^{15}$N-TROSY-HSQC NMR. Increasing amounts of detergent induce changes in the overall fold and general changes across the entire NMR spectrum (especially in the Trp region) in the major population of the Bcl-2 ΔTM protein. Refolding Bcl-2 ΔTM upon purification directly into DPC micelles results in identical spectral behavior for the entire population, as also observed for intact, fully Bcl-2 protein in DPC micelles (Supplementary Figs. 7 and 8). The detergent environment causes major conformational changes towards an active protein state, as demonstrated by successfully binding Bax-derived BH3 peptides to both the full-length Bcl-2 and the Bcl-2 ΔTM proteins in DPC micelles (Supplementary Figs. 3, 9).

This transition behavior reflects the propensity of other Bcl-2 proteins, such as Bcl-x_L and Bcl-w to adopt structures suitable for functioning in a membrane. Following the addition of the membrane-mimicking detergent DPC, also their N-terminal head domain with its common globular fold adopted such structures[45–47]. This change in the globular fold was clearly visible in all the $^1$H/$^{15}$N NMR spectra acquired in the cited studies, especially in contributions of the Trp residues, which reflect dramatic changes in their local environment. Presumably, full-length Bcl-2 is insoluble due to its hydrophobic TM domain extended away from the main protein body and not hidden in its BH3 binding groove region as in other Bcl-2 proteins which can exist as soluble cytosolic subpopulations[16].

As can be seen in the solution NMR spectra (Fig. 4), the main N-terminal head domain (a common globular fold) of the entire Bcl-2 protein becomes dynamically restricted in detergent micelles while the FLD region remains highly flexible in the functional state where this loop region successfully binds the Nur-77 peptide (Supplementary Fig. 4). Our observations show that this arrangement of full-length Bcl-2 is preserved following the replacement of its micellar environment by native-like lipid membranes. NR showed that the protein body was buried in the membrane and remained dynamically restricted (as shown by

solid-state NMR), while the loop region remained flexible outside the membrane. Various residues and their positions could be identified from solution NMR-based residue assignment in the Bcl-2/micelle systems (Supplementary Fig. 5), indicating that the protein adopted similar conformations in both membrane-mimicking environments. While NMR experiments (Supplementary Fig. 4) showed a clear binding of the Nur77 peptide to the FLD, this binding event did not induce any specific structural features into this intrinsically disordered extended loop and the remaining protein body was not affected. This indicates that the regulative loop region can act in a spatially decoupled manner from the main protein core with its BH3-BH1 groove region. Presumably, as proposed for Bcl-x$_L$ inhibition of Bax[48], the loop provides the flexibility for the helices α1 from both proteins to form a second binding interface in addition to the main Bcl-x$_L$ groove interface which sequesters BH3 helix of Bax. We found that the full-length Bcl-2 protein resides in host membranes in a fully embedded conformation as a multidomain spanning integral membrane protein. This not only reflects the conformational plasticity required for proper functioning[15], but also the location required for blocking cell-killing Bcl-2 family members like Bax. Cytosolic Bax proteins can translocate to the MOM where they partially penetrate and form oligomeric membrane pores, thereby releasing apoptotic factors and ensuring cellular death[34,49]. Bcl-2 is thought to block membrane perforation by Bax (and presumably its apoptotic relatives) by engaging with the BH3 domain, after partial membrane penetration, thus sequestering the apoptotic protein-membrane boundary before any dimerization and subsequent membrane damage occurs[50].

To restrain apoptotic Bcl-2 proteins, the Bcl-2 protein itself has an extended groove interface formed by its BH3, BH1, and BH2 domains. By engaging four complementary pockets of the groove region, the Bcl-2 protein is binding four or more hydrophobic residues on consecutive helical turns of these BH3 motifs of apoptotic Bcl-2 proteins[31]. This design enables strong engagement with variable BH3 motifs from different apoptotic multidomain proteins but also from BH3 only proteins; something clearly confirmed by our NMR titration studies using Bax-BH3 and Bim-BH3 peptides and various Bcl-2 constructs where the BH1-BH3 groove region was conserved (Supplementary Fig. 3, 9, 10 and 11). In all cases, the presence of BH3 peptides caused significant CSPs for residues presumably in direct contact with those peptides in the groove region. In addition, the entire protein experienced a structural adaption in response to the bound ligands, as visible in more moderate CSPs across most protein residues. This type or adaption ("structural breathing") is most likely the reason why the Bcl-2 protein can bind various BH3 motifs of diverse Bcl-2 family members with high affinity, even if mutations in this binding pocket occur which dramatically reduce the affinity of rigid cancer drugs like venetoclax targeting this groove region[31]. The hydrophobic groove region of Bcl-2 is central for blocking cell death and can most likely exert direct recognition with high affinity of varying BH3 death domains of apoptotic proteins via a binding interface facilitated by structural flexibility.

The membrane environment plays a crucial role by changing proteins' local abundances and their affinities through two-dimensional crowding, while keeping the sequestration spatially separate from the FLD region, which regulates Bcl-2's conformation and activities by bundling numerous cytosolic signals[51]. In this manner, Bcl-2 protects healthy cells by sequestrating any transiently membrane-bound Bax at the MOM[52]. Many tumors escape therapy-induced death (often via activation of apoptotic Bcl-2 family members) by upregulation of this Bcl-2 protein and its cell-protecting relatives[12,53], all of which presumably function in a similar manner to Bcl-2. Our model of Bcl-2 fully embedded

in its host membrane provides the molecular basis for the protein's cell-protecting activity and may facilitate the discovery of drugs targeting currently non-treatable tumors.

## Methods

**Preparation of isotopically-labeled Bcl-2 proteins.** Full-length uniformly $^{15}$N and $^{15}$N/$^{13}$C labeled Bcl-2 was expressed and purified following our recent protocol[32]. Bcl-2 was deuterated (>90%) as follows: M9 media was prepared by adding 13 g KH$_2$PO$_4$, 10 g K$_2$HPO$_4$, 9 g Na$_2$HPO$_4$, 2.4 g K$_2$SO$_4$, 2.5 ml of MgCl$_2$ (2.5 M stock), 1 ml thiamine (30 mg/ml stock), 2 g glucose (non-deuterated), and 2 g NH$_4$Cl (non-deuterated) per liter of milli-Q water. The medium was supplemented with trace elements (1 ml of 50 mM FeCl$_3$, 20 mM CaCl$_2$, 10 mM MnCl$_2$, 10 mM ZnSO$_4$, 2 mM CoCl$_2$, 2 mM CuCl$_2$, 2 mM NiCl$_2$, 2 mM Na$_2$MoO$_4$ and 2 mM H$_3$BO$_3$ per liter), 100 μg/ml carbencillin and 34 μg/ml chloramphenicol. The pH of the medium was adjusted to 6.9 and sterile filtered prior to use.

A glycerol stock of transformed BL21(DE3) Rosetta™ cells with Bcl-2 encoded in a pET-15b vector (Novagen) was ordered from GenScript (Leiden, Netherlands). The cells were cultured in media with step-wise increases in the concentration of deuterium oxide (Armar Isotopes GmbH, Germany). Initially, cells from the glycerol stock were grown in 20 ml of LB media (10 g Bacto tryptone, 5 g yeast extract, 5 g NaCl per liter, pH adjusted to 7.5, and sterile filtered) overnight at 37 °C. The following day 500 μl of the resulting culture was transferred into M9 media prepared as above, but included 50% D$_2$O. Cells were grown at 37 °C for an additional day, then 3 ml of the resulting culture was collected and centrifuged at 4400 × g for 10 mins. The obtained pellet was transferred into M9 medium containing 75% D$_2$O, and the procedure was repeated, with step-wise increases to 90% and finally 99.8% D$_2$O. The cells grown in 90% and 99.8% D$_2$O were left to incubate for an extra day at 37 °C before harvesting, due to their slower growth rate. The final batch for deuteration was prepared by transferring 100 ml of the adapted culture to 900 ml of M9 medium in 99.8% D$_2$O. The cells were cultivated at 37 °C, induced by adding 1 mM IPTG upon reaching OD$_{600}$ ~ 0.6, and further grown overnight. The following day deuterated Bcl-2 protein was harvested by centrifuging the culture at 4400 × g for 30 min, then resuspending the cells in 20 ml 50 mM Tris, pH 8.0, and freezing them at −80 °C until purification. Bcl-2 protein was labeled with either $^{13}$C or $^{15}$N using the same routines, but replacing the glucose or NH$_4$Cl with the same amount of isotope-enriched variants.

**Preparation of truncated Bcl-2 variants.** Primers for constructing the Bcl-2 ΔTM truncation were ordered from Eurofins Genomics (Ebersberg, Germany), which introduces a stop codon to terminate the protein at position 207, thus disabling the membrane-binding region of the protein. Site-directed mutagenesis was performed using the Quikchange approach (Stratagene) with oligonucleotide forward primer 5′-GGACCTTCGATGCGTTAACTGTTTGATTTTTCG-3′ and reverse primer 5′-CGAAAAATCAAACAGTTAACGCATCGAAGGTCC-3′, respectively. Proper insertion of the stop codon TAA was verified by DNA sequencing (Eurofins Genomics).The Bcl-2 truncated variant Bcl-2 ΔN(1-82) was purchased from GenScript® (Leiden, Netherlands) and sub-cloned into a pET-15b expression vector (Novagen). The construct is aimed to delete the first 82 N-terminal amino acids, starting the N-terminus with residues MGP, while it maintains the C-terminal residues. Worth noting that all Bcl-2 variants carry an additional three N-terminal residues, GSH, due to the design of the pET-15b expression vector. Expression and purification of both mutants were carried out in an identical manner as for the full-length protein and the Bcl-2 ΔTM mutant. Soluble Bcl-2 ΔTM protein was purified directly from the minor soluble fraction after sonication and centrifugation, and then further purified using identical procedures. For concentration measurements, $\varepsilon_{280}$ = 44920 M$^{-1}$ cm$^{-1}$ was used for full-length Bcl-2, for the mutant Bcl-2 ΔTM, $\varepsilon_{280}$ = 37930 M$^{-1}$ cm$^{-1}$, and for the Bcl-2 ΔN(1-82) mutant $\varepsilon_{280}$ = 33460 M$^{-1}$ cm$^{-1}$, respectively.

**In vitro binding assays for Bcl-2 proteins.** The functionality of the Bcl-2 protein in both Brij-35 and DPC micelles was verified as recently described[32]. We checked the full-length protein's functionality in these environments by monitoring chemical shift perturbations using $^{15}$N TROSY-HSQC NMR (Supplementary Figs. 3, 4) for two purposes. First, to determine the affinity of Bax-BH3 peptide (which is known to bind Bcl-2[32] and mimics the cell-death inducing Bcl-2 homology domain 3, BH3 of Bax) to the hydrophobic groove of Bcl-2[38]. Second, to determine the binding and affinity of the Nur77 peptide, which is known to bind to Bcl-2's loop region and trigger Bcl-2's activities[35], to check the functionality of the regulatory loop region (residues 32–90 here), which are removed in truncated soluble Bcl-2 variants[29,30]. For the Bax-BH3 peptide binding assay, the 36-mer Bax-BH3 peptide from *Mus musculus* (Ac-QPPQDASTKKLSECLRRIGDELDSNMELQRMIADVD-NH$_2$) was ordered from GenScript (Leiden, Netherlands) and dissolved in NMR buffer (5 mM DPC micelles, 20 mM NaPi, 20 mM NaCl, 2 mM TCEP at pH 6.0), and added in a 1:1, 1:3, 1:6 and 1:12 molar ratio to 0.3 mM Bcl-2 and 0.25 mM Bcl-2 ΔTM samples respectively, followed by obtaining $^1$H-$^{15}$N-TROSY-HSQC spectra with and without peptide added. All spectra for Bcl-2 were measured on an 850 MHz magnet while for Bcl-2 ΔTM spectra were measured on a 600 MHz

magnet, at 310 K. Chemical shift perturbations due to added peptide are displayed in Supplementary Fig. 3.

For the Nur77 binding assay, the Nur77 peptide (Ac-FSRSLHSLL-NH₂) was ordered from GenScript (Leiden, Netherlands) and dissolved in NMR buffer (5 mM DPC micelles, 20 mM NaPi, 20 mM NaCl, 2 mM TCEP at pH 6.0), pH adjusted and added in a 1:6, 1:12 and 1:24 molar ratio to 0.3 mM Bcl-2 samples as described above, followed by obtaining ¹H-¹⁵N-TROSY-HSQC spectra with and without peptide added. Chemical shift perturbations due to added peptide are displayed in Supplementary Fig. 4.

For the Bim-BH3 peptide binding assay, the 36-mer Bim (138-174) BH3 peptide from *Homo sapiens* (Ac-EPADMRPEIWIAQELRRIGDEFNAYYAR RVFLNNYQA-NH₂) was ordered from GenScript® (Leiden, Netherlands), and dissolved in NMR buffer (5 mM DPC micelles, 20 mM NaPi, 20 mM NaCl, 2 mM TCEP at pH 6.0), and added in a 1:1, 1:3 and 1:6 molar ratio to 0.4 mM Bcl-2ΔN(1-82), followed by obtaining ¹H-¹⁵N-TROSY-HSQC spectra with and without the peptide added. Bim peptide was forming a colloidal suspension in buffer, and upon addition of the peptide to the protein at respective molar ratios, NMR spectra were recorded on clear samples upon centrifugation (Supplementary Fig. 11).

**Reconstitution of Bcl-2 protein in lipid bilayers**. Bcl-2 protein was reconstituted using a protocol developed by us[21], with some modifications as follows: A stock suspension of DMPC vesicles was prepared by adding 5 mM DMPC to 20 mM NaPi, 50 mM NaCl, 2 mM DTT, 1 mM EDTA at pH 7.4, then sonicated using a ultrasonic cleaner (VWR, USA). Mixed micelles were prepared by solubilizing DMPC or DMPC-d₆₇ vesicles in 0.05 % (w/v) Brij-35 (Sigma-Aldrich), together with a stock solution containing pure detergent-solubilized Bcl-2 protein until the solution became clear. The concentration of DMPC lipids and protein was tuned to generate a 1:70 (mol/mol) protein:lipid ratio. Brij-35 in the proteoliposome mixture was removed by adding approximately 1 ml of Biobeads™ (Biorad) to 10 ml of protein solution, and slowly set to mix on a rocking table at 4 °C overnight. In the following days, the Biobeads™ were removed and this process was repeated until a cloudy solution emerged. Proteoliposomes were purified using a 25% and 40% (w/v) sucrose gradient, prepared by dissolving sucrose in 20 mM NaPi, 50 mM NaCl, 1 mM EDTA, pH 7.4. The sample was centrifuged at 75000 × g for 60 min at 4 °C using a SW-41 Ti rotor together with an Optima L 90 K Ultracentrifuge (Beckman Coulter, USA). Isolated proteoliposomes were further purified by adding buffer and centrifugation using a Micro Star 17 tabletop centrifuge (VWR, USA) for 20 mins x2, and the sample was subsequently used in solid-state NMR or neutron scattering experiments. Analysis of the proteoliposome pellet by SDS-PAGE confirmed the presence of Bcl-2 protein with the correct molecular size. Fully deuterated DMPC and DMPC-d₆₇ were purchased from Avanti Polar Lipids (Alabaster, USA).

**Neutron reflectometry measurements of silicon-supported membranes**. Silicon single crystals (50 × 80 × 15 mm; Sil'tronix, France) with a 80 × 50 mm (111) face polished to 3 Å rms roughness were assembled into purpose-built solid/liquid flow cells with the polished face of the crystal in contact with the liquid reservoir of the liquid trough within the cell.

NR measurements were acquired with the SURF reflectometer (which uses a white beam of neutrons with wavelengths ranging from 0.5 to 7 Å) at the STFC ISIS Pulsed Neutron and Muon Source, Rutherford Appleton Laboratory, Oxfordshire, UK. The silicon crystal-containing solid/liquid flow cells were placed in the sample position in the instrument and connected to a L7100 high-performance liquid chromatography pump (Merck, Hitachi), for changing the H₂O/D₂O ratio of the solution in the cell. The following ratios were used: 100%, 80%, 38%, 0% D₂O/H₂O buffer (20 mM NaPi, 50 mM NaCl, 1 mM EDTA, pH 7.4).

In specular NR experiments, intensity is measured as a function of the angle and/or wavelength of the beam relative to the sample surface (46, 47), expressed as the momentum transfer vector, $Q_z = (4\pi \sin \theta)/\lambda$, where $\lambda$ is wavelength and $\theta$ is the incident angle). The white beam instruments can probe a wide range of $Q_z$ space at a single angle of reflection due to the use of a broad neutron spectrum. To obtain reflectivity data across a $Q_z$ range of ~0.01 to 0.3 Å⁻¹, glancing angles of 0.35°, 0.65°, and 1.5° were used.

During the experiments, the bare silicon/water interface was initially examined before the Bcl-2/DMPC vesicles were deposited by vesicle rupture onto the silicon interface. The composite interfacial structure was then examined under multiple solution isotopic contrast conditions so the relative distributions of both the lipid and protein components within the membranes could be resolved by simultaneous analysis of all the contrasts.

**Neutron reflectometry data analysis**. The magnitude of the coherent neutron scattering length of nuclei varies randomly across the periodic table[54]. Some isotopes, most usefully the hydrogen isotopes protium (99.98% natural abundance) and deuterium (0.015% natural abundance), have different neutron scattering lengths. Differential hydrogen isotope labeling of samples for neutron scattering experiments is commonly used to collect a series of datasets pertaining to samples under chemically similar but isotopically different conditions. Isotopic labeling of the bulk solution and sample to produce multiple scattering 'contrasts' is advantageous when analyzing complex structures. This is because multiple datasets strongly limit the number of potential structural solutions when they are

simultaneously analyzed and allow resolution of distributions of individual components within analyzed complexes if a suitable labeling strategy is employed[55].

NR data were analyzed using RasCal software (A. Hughes, ISIS Spallation Neutron Source, Rutherford Appleton Laboratory), which fits layer models describing the interfacial structure calculated using Abelès matrix calculation[33] to experimental data. In this approach, the interface is described as a series of slabs, each characterized by its nSLD, thickness and roughness. The reflectivity for the model's starting point is then calculated and compared with the experimental data.

The final datasets consisted of four individual reflectivity profiles obtained from the membrane-coated silicon/water interface collected in 100% D₂O, 80% D₂O and 20% H₂O, silicon-matched water (Si-MW, 38% D₂O and 62% H₂O) and 100% H₂O. The membrane was composed of hydrogenous (i.e. natural abundance hydrogen) DMPC and deuterated (d-)Bcl-2 protein. The difference in isotope labeling of the membrane constituents allows the protein and lipid components to be highlighted in different solution contrasts. Specifically, the D₂O solution contrast was sensitive to the structure of the lipid component of the membrane due to the large difference in SLD (ρ) between the lipid and D₂O (Table 1). Similarly, the H₂O solution isotopic contrast was sensitive to the protein component of the membrane due to the strong difference between these two components.

For modeling the interfacial structure, the known scattering length densities of the protein and lipid components were used as constraints and the membrane was modeled as a five-layer structure. The layers of the interfacial structure (from silicon to solution) were: SiO₂, the inner headgroup region, bilayer hydrophobic tail region and outer headgroup region. These layers were fitted with the same SLD, thickness and roughness across all four solution isotopic contrasts. The membrane structures at the silicon surface (and immersed in the bulk solution) were fitted with varying SLDs between contrasts to account for the presence of water within the layers and labile hydrogen exchange in different solution isotope mixtures. Interpretation of the interfacial structure from the resulting SLD profile informed further fitting. The positions of each component in the protein/lipid membrane were linked with the known SLDs of the components and the solution isotopic contrast to determine the volume fractions of the protein and lipid components within the model membrane[55].

**Solid-state NMR spectroscopy analysis of full-length Bcl-2 in proteoliposomes**. All ¹H and ¹³C MAS NMR spectra were acquired using a 3.2 mm HCN MAS Bruker probe at an 850 MHz Avance III HD spectrometer (Bruker, Switzerland) and proteoliposomes in NMR buffer (20 mM NaPi, 50 mM NaCl, 1 mM EDTA, 1 mM DTT, pH 7.4). ¹H MAS NMR spectra were acquired using a single π/2- pulse with a 3.1 µs (¹H) duration, and 1.5 s repetition delay and a sample spinning rate of 15 kHz. One-dimensional ¹³C solid-state CP and INEPT NMR spectra were acquired to obtain information on both the rigid and more flexible parts of the protein[24].

The CP NMR experiments were performed at varying temperatures with a contact time of 0.5 ms, an RF field of 50 kHz at the ¹³C frequency and a ¹H ramp power from 34 to 69 kHz, and 63 kHz of decoupling power. For refocused INEPT and direct polarization experiments, the ¹³C pulse duration was typically 3.35 µs while the decoupling lowered for INEPT (57 kHz swft ppm) while the acquisition time increased from 19 ms to 37 ms. The ¹³C chemical shift of 38.5 ppm for adamantane was used as an external reference. In two-dimensional NMR experiments, ¹³C CP MAS NMR spectra were acquired using a π/2-pulse with 3.1 µs (¹H) duration followed by ¹H ramped CP at ¹³C (79 kHz; 40-81 kHz ramp) for 1 ms, a ¹H SPINAL-64 decoupling at 81 kHz, and a 1.5 s repetition delay at a sample spinning rate of 15 kHz. 2D ¹³C-¹³C DARR NMR experiments were carried out with 25, 50, and 100 ms mixing times using a 15 kHz MAS rate.

**Solution state NMR spectroscopy of full-length Bcl-2 protein in micelles**. ¹H-¹⁵N TROSY experiments were performed using 0.3 mM ¹³C,¹⁵N-labeled full-length Bcl-2 in 5 mM DPC micelles (20 mM NaPi, 20 mM NaCl, 2 mM TCEP at pH 6.0) and 328 K for backbone assignments, and ¹⁵N-labeled 0.3 mM and 0.25 mM Bcl-2 ΔTM in 5 mM DPC micelles (20 mM NaPi, 20 mM NaCl, 2 mM TCEP at pH 6.0) and 310 K for Bax-peptide titration experiments. ¹H-¹⁵N TROSY spectra using a ¹⁵N T₂ relaxation filter of 410 ms CPMG delay and relaxation delay of 2 s were acquired at 310 K under similar buffer conditions. All 2D experiments were performed with 8 or 16 scans, time-domain sizes of 256 (¹⁵N) × 2048 (¹H) complex points and sweep widths of 11029.412 Hz and 2412.313 Hz along the ¹H and ¹⁵N dimensions, respectively. Resonance assignments of ¹³C- and ¹⁵N- labeled full-length Bcl-2 in DPC micelles were obtained from INEPT (insensitive nucleus enhancement polarization transfer), BEST-TROSY (Band-selective Excitation Short-Transient-Transverse Relaxation Optimized Spectroscopy) type triple-resonance experiments: HNCACB, HN(CO)CACB, HNCA, HN(CO)CA, HNCO and HN(CA)CO. The pulse programs were obtained from the Bruker TopSpin 3.6.1 library. These data were processed with free NMRPipe and NMRDraw softwares[56], and visualized and analyzed with the programs CCPN2.1.5[57] and Sparky[58]. All NMR measurements for backbone chemical shift assignments were acquired at 328 K using an Avance 850-MHz NMR machine with a triple-resonance cryogenic probe (Bruker, Germany).

**Statistics and reproducibility**. The structure of the Bcl-2:DMPC complex was examined in triplicate by NR. Twice using a labeling strategy of d-Bcl-2:h-DMPC (Fig. 2) and once with h-Bcl-2:d-DMPC (Supplementary Fig. 2). Model parameter error estimation from the fits of these NR datasets was conducted using Rascals Bayesian Error estimation routines, with the log-likelihood function described in terms of chi-squared[22]. Marginalized posteriors were obtained using a Delayed Rejection Adaptive Metropolis algorithm[22], and the best-fit parameters taken as the distribution maxima; the uncertainties presented here are from the shortest 95% confidence intervals of each distribution.

**Reporting summary**. Further information on research design is available in the Nature Research Reporting Summary linked to this article.

## Data availability

The authors declare that all data supporting the findings of this study are available within the article, its supplementary information file and from the corresponding authors upon reasonable request. Neutron reflectometry data is directly available from 10.5286/ISIS.E.94115547 and 10.5286/ISIS.E.RB1720424 for experiment RB1810627 and RB1720424 respectively. Models used in NR analysis are available upon request.

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

## Acknowledgements
G.G. acknowledges financial support from the Swedish Research Council, the Swedish Cancer Foundation, the Kempe Foundation, the Knut and Alice Wallenberg Foundation ('NMR for Life' Programme), the SciLifeLab, the Swedish National NMR Centre and Umeå Insamlingsstiftelse. This work was supported by ISIS beam time Awards RB1720424 and RB1810627.

## Author contributions
G.G., L.A.C., and B.H.W.-K. conceived the project. J.Å. expressed and purified proteins including biophysical characterization. A.U.M., J.Å., T.S., C.P. performed NMR studies. G.G., J.Å., L.A.C., B.H.W.-K., M.C., A.P.G.D., and T.S. performed NR experiments. A.U.M., L.A.C., B.H.W.-K., C.P., and G.G. analyzed the data; G.G., J.Å., L.A.C., and B.H.W.-K. wrote the paper. G.G. supervised the overall project.

## Funding

## Competing interests
The authors declare no competing interests.
