## [Peer Review File · Communications Biology]

Reviewers' comments:

Reviewer #1 (Remarks to the Author):

6490-0

The molecular basis of cell protection by Bcl-2 proteins

Ameeq Ul Mushtaq^{1,a}, Jörgen Ådén^{1,a}, Luke A. Clifton², Hanna Wacklin-Knecht^{3,4}, Mario Campana², Artur P. G. Dingeldein¹, Cecilia Persson⁵, Tobias Sparrman¹ and Gerhard Gröbner^{1,*}

This is an interesting paper from a group of world-class membrane biologists that determines the structure of Bcl-2 in a membrane environment. A series of well-executed experiments evaluating Bcl-2 structure by neutron reflectometry and solid-state NMR spectroscopy are presented that convincingly demonstrate that Bcl-2 is entirely buried within the membrane with only the flexible loop domain exposed. The strength of this combined approach is that neutron reflectometry (NR) provides an assessment of the distribution and therefore location across a lipid membrane while solid state NMR spectroscopy provides structural and dynamic information at atomic resolution at near-physiologic conditions. NR-membrane profiles clearly demonstrate that Bcl-2 resides predominantly within the membrane's hydrophobic interior, with no protein density outside the membrane. Solid-state NMR demonstrates that the majority of Bcl-2 residues are dynamically restricted. In addition, the flexible loop domain is shown to be flexible using ¹⁵N T2 relaxation experiments.

The Bcl-2 family exerts its cell death effects at the mitochondrial outer membrane, thus an understanding of the structure of these proteins within a membrane is highly significant and represents an advance for the field. Nevertheless, as presented, the study evaluates primarily Bcl-2. Several experiments evaluating impact of Bax BH3 peptides and Nur77 BH3 peptides on the structure of Bcl-2 are included in the supplement, but the functional implications for these structural changes are not well-discussed in either the results or discussion section, thus conclusions regarding the molecular basis of cell protection are limited.

Specific comments:

1. The abstract is not clearly written. The novel insights into Bcl-2 structure within a membrane are not well-described. The last three sentences are speculative.
2. The first full paragraph on page 4 has a typographical error (extert)
3. The introduction is nicely written, and Figure 1 provides a nice schematic diagram of the contributions of the two techniques used.
4. Supplementary Figure 2: Bcl-2 is mis-spelled
5. Supplementary Figure 3 is not clearly discussed in the text. This figure presents data that addresses how Bcl-2 structure changes upon addition of Bax -BH3, which is quite relevant for the molecular mechanism, but the functional implications of these structural changes are not well- discussed.
6. Supplementary Figure 4 is not clearly discussed in the results section of the text. The reason for the choice of Nur77 BH3 peptide is not apparent until the discussion. The manuscript would be improved by introducing Nur77 binding to Bcl-2 earlier.
7. The paper would be improved by inclusion of other well-studied BH3 peptides (activators and sensitizers), to more clearly define how Bcl-2 structure in the membrane may impact its function in cell death.

Reviewer #2 (Remarks to the Author):

The paper makes use of a full length version of Bcl-2.

This paper is a significant advance upon previous studies where the proteins were usually truncated at the C-terminus. The protein is then subjected to analysis by neutron reflectometry and solid state NMR. The methods employed of a high technical quality and the analysis are for the most part sufficient and correct. The conclusion is that the Bcl-2 protein embeds itself into the membrane more deeply than previously proposed. This is suggested by the model of the neutron reflectivity data in which the protein is found at the level of the hydrophobic lipid tails and by the NMR analysis which shows a conformationally restricted protein with only the FLD as a flexible region. For the most part the data is convincing and will be a significant addition to the literature on these important proteins. There are however some points which need to be clarified and these are listed below.

Major points .

The results of the neutron reflectometry suggest that the protein enters fully into the core of the membrane showing approximately 9% of the membrane being protein. Compared to the protein-free membrane the water content declines from an already very low 2% to 1%. Whilst these values presumably are within experimental error the same, it is difficult to reconcile this with the insertion of a water soluble protein into the dry core of the membrane. There is a discussion about the conformational rearrangement required for a protein to convert from a water soluble form to a membrane embedded form but this is too brief to explain to the reader the author's conclusions as to the crucial event which is being characterised here. Firstly, the language needs to be tightened up for example Line 308 "Generally, the N-terminal protein heads of these Bcl-2 members are amphitropic (soluble or membrane-bound)". Amphitropic describes the affinity for both water and membranes not the state of being bound, and what is a N-terminal protein head?. The critical event here is that the amphitropic helices will need to rotate to expose their hydrophobic sides to the membrane rather than the protein core and the hydrophilic sides will now form a hydrophilic core. This might be expected to increase the water content of the membrane unless a highly defined new set of hydrogen bonding interactions takes place to satisfy the polar interactions of all the hydrophilic side chains. In most previous cases of proteins inserting into the membrane like this the resulting confirmation of the altered membrane embedded form has not created a rigid dry structure in the membrane but by most measures has been flexible and has introduced water into the membrane core. In fact the membrane embedded form of BCL-2 has even introduced channels in the lipid bilayer PNAS May 13, 1997 94 (10) 5113-5118, Biophys J Volume: 104 Issue: 2 Pages: 421-431 (although not as clear as the oligomeric Bax). The structure being presented here is therefore a novel form of membrane embedded protein and should therefore be discussed in clearer and more precise detail as the authors explanation of this truly revolutionary structure.

Minor points figure 2

Figure 2 What is the 4th solution condition in figure 2b with Bcl2 also please explain the flat base to the data fit in the pure DMPC data.

The description of the data fitting in Results & Materials and methods is confusing . In the first "The analysis was carried out using Abelès matrix formalism, which divides the focal (why focal?) membrane into discrete homogeneous layers across the bulk interface". In the second "NR data were analyzed using RasCal software..., which fits layer models describing the interfacial structure calculated using Parratt's recursive formalism to experimental data. In this approach, the interface is described as a series of slabs, each characterized by its nSLD, thickness and roughness". Is this the same approach?

General Comments

1. We have removed misspellings, typos etc in the revMS. And in Figure 2 we have included a missing label (80% D₂O).
2. In revMS we have removed old reference 50 (wrong one), cited there the correct one (ref. 33) and included various new references (in red) including the ones suggested by reviewers.
3. In supplementary Information, we have fixed some minor layout problems (Fig. axis e.g label orientation) and improved figure captions (in red) when necessary.
4. In Supplementary Information we have modified following Figures and Table:
S5: updated with better partial assignments of Bcl-2 residues with focus on loop region and N-terminus (helix α 1 region).
S6: updated figure with the main figure now more clear by removing one of the overlaid spectra (but still included in insert box (right) spectral region, where now also an additional experiment is included).
Supplementary Table 1: included now SLD values for h-Bcl-2 in D₂O/H₂O (red).
5. In Supplementary Information we have included following new Figures:
S10: Titration of Bax-BH3 peptide against the Bcl-2 Δ N(1-82) protein variant.
S11: Titration of BIM-BH3 peptide against the Bcl-2 Δ N(1-82) protein variant.
6. In Supplementary Information we have now included a supplementary Method section. This includes now i) description of solution NMR experiments and their analysis (including references); ii) functional binding assays for BIM-BH3 peptide; and iii) preparation of Bcl-2 protein variants.

Specific comments to both reviewers

Reviewer 1

This is an interesting paper from a group of world-class membrane biologists that determines the structure of Bcl-2 in a membrane environment. A series of well-executed experiments evaluating Bcl-2 structure by neutron reflectometry and solid-state NMR spectroscopy are presented that convincingly demonstrate that Bcl-2 is entirely buried within the membrane with only the flexible loop domain exposed. The strength of this combined approach is that neutron reflectometry (NR) provides an assessment of the distribution and therefore location across a lipid membrane while solid state NMR spectroscopy provides structural and dynamic information at atomic resolution at near-physiologic conditions. NR-membrane profiles clearly demonstrate that Bcl-2 resides predominantly within the membrane's hydrophobic interior, with no protein density outside the membrane. Solid-state NMR demonstrates that the majority of Bcl-2 residues are dynamically restricted. In addition, the flexible loop domain is shown to be flexible using ^{15}N T2 relaxation experiments.

The Bcl-2 family exerts its cell death effects at the mitochondrial outer membrane, thus an understanding of the structure of these proteins within a membrane is highly significant and represents an advance for the field. Nevertheless, as presented, the study evaluates primarily Bcl-2. Several experiments evaluating impact of Bax BH3 peptides and Nur77 BH3 peptides on the structure of Bcl-2 are included in the supplement, but the functional implications for these structural changes are not well-discussed in either the results or discussion section, thus conclusions regarding the molecular basis of cell protection are limited.

Our reply: Our binding studies of Bax BH3 peptide (hydrophobic groove region) and Nur 77 peptide (extended loop region) were originally performed to confirm that the Bcl-2 protein used by us is fully functional. Upon the valuable suggestion by the reviewer we have to exploited the potential of those data with focus on Bcl-2 molecular functioning at the membrane level:

- *We have now highlighted more prominently those binding studies in the MS (s. lines 257-271 in revMS) and presented them now earlier in the revMS (as suggested by both reviewers). We have even expanded those binding studies using various Bcl-2 constructs to highlight the importance of its extended groove region for recognizing BH3 motifs of apoptotic proteins (s. Fig. S3, S9 with lines 404-414 in revMS; and together with new Figs. S10, S11 and lines 463-481 in revMS). In detail:*
 - o *As suggested under point 7, we carried out an additional titration using the BH3 binding domain of a BH3 only protein BIM (s. new Fig. SI 11). The observed changes were very similar to the one obtained for the Bax BH3 peptide (SI 10) using the same Bcl-2 $\Delta\text{N}(1-82)$ construct. Included now in main text (s. lines 467-471 in revMS).*
 - o *Comparing Bax BH3 peptide binding to this Bcl-2 $\Delta\text{N}(1-82)$ construct with it flexible loop region missing, revealed that the main protein fold is conserved as in intact Bcl-2 (Fig. SI3) with Bax BH3 peptide inducing similar changes as seen in Fig. SI10).*

- *We discuss now in more detail the functional implications of these peptide binding events on Bcl-2's molecular mechanism, based on structural changes observed upon BH3 and NUR 77 peptide binding; and discuss them critically also with respect to earlier work using truncated Bcl-2 variants and Bcl-xL relatives (s. lines 443-451 and again 463-481 and ref. in revMS).*

Specific comments:

1. The abstract is not clearly written. The novel insights into Bcl-2 structure within a membrane are not well-described. The last three sentences are speculative.

Our reply: We have now modified the abstract and the novel insights into Bcl-2 structure are better highlighted now. The last three sentences are changed to be less speculative in promoting a molecular mechanistic model for Bcl-2 in the membrane environment (s. abstract in revMS).

2. The first full paragraph on page 4 has a typographical error (extert)

Our reply: Fixed now.

3. The introduction is nicely written, and Figure 1 provides a nice schematic diagram of the contributions of the two techniques used.

Our reply: We fully agree.

4. Supplementary Figure 2: Bcl-2 is mis-spelled

Our reply: Fixed now in revised Supplements.

5. Supplementary Figure 3 is not clearly discussed in the text. This figure presents data that addresses how Bcl-2 structure changes upon addition of Bax -BH3, which is quite relevant for the molecular mechanism, but the functional implications of these structural changes are not well-discussed.

Our reply: As mentioned in our reply to general comments above, we have discussed in more detail Bcl-2 structural changes upon binding of BH3 domains to its hydrophobic groove (s. lines 257-271 and 463-481 in revMS).

6. Supplementary Figure 4 is not clearly discussed in the results section of the text. The reason for the choice of Nur77 BH3 peptide is not apparent until the discussion. The manuscript would be improved by introducing Nur77 binding to Bcl-2 earlier.

Our reply: As mentioned above we introduced Nur77 binding now earlier, and have discussed the Nur77 peptide binding in more detail in result and discussion section (s. lines 257-271 and 443-451 in revMS).

7. The paper would be improved by inclusion of other well-studied BH3 peptides (activators and sensitizers), to more clearly define how Bcl-2 structure in the membrane may impact its function in cell death.

Our reply: As mentioned above we have now included additional experiments using the BIM BH3 peptide (s. new supplementary Fig. S11 and lines 467-471 in the 463-481 context in revMS).

-Reviewer 2

The paper makes use of a full length version of Bcl-2.

This paper is a significant advance upon previous studies where the proteins were usually truncated at the C-terminus. The protein is then subjected to analysis by neutron reflectometry and solid state NMR. The methods employed of a high technical quality and the analysis are for the most part sufficient and correct. The conclusion is that the Bcl-2 protein embeds itself into the membrane more deeply than previously proposed. This is suggested by the model of the neutron reflectivity data in which the protein is found at the level of the hydrophobic lipid tails and by the NMR analysis which shows a conformationally restricted protein with only the FLD as a flexible region. For the most part the data is convincing and will be a significant addition to the literature on these important proteins. There are however some points which need to be clarified and these are listed below.

Major points .

1. The results of the neutron reflectometry suggest that the protein enters fully into the core of the membrane showing approximately 9% of the membrane being protein. Compared to the protein-free membrane the water content declines from an already very low 2% to 1%. Whilst these values presumably are within experimental error the same, it is difficult to reconcile this with the insertion of a water soluble protein into the dry core of the membrane.

Our reply: We entirely agree with the reviewer's comment about soluble proteins here. In general water soluble proteins have significant associated water which is easily observable coincidentally with the protein density within the membrane, if present. However, the full length Bcl-2 used in our experiments is not water soluble, and is natively located on/in the mitochondrial membrane. As opposed to many other experiments that were done using C-terminally truncated, soluble versions of Bcl-2, the full length protein was reconstituted into the supported lipid bilayers by direct fusion of proteo-liposomes (as described in Methods section on page 29 in revMS). Our fits suggested that little measurable water was found coincidentally with the protein in the hydrophobic core of the membrane, which is also consistent with it being almost completely embedded in the membrane. This observation came from using the minimal number of assumptions in the model-to-data fitting of the experimental reflectivity data. In our approach the volume fractions of the protein, lipid and water components of the membrane can vary between 0-100% and have large positional freedom (can be located anywhere across the membrane), but are constrained to maintain the molecular volumes and lipid headgroup to tail stoichiometry. Using this approach we found Bcl-2 was located in the hydrophobic core of the membrane (which was initially unexpected) and that it was low in measurable hydration in this environment. This observation was seen across three independent samples (run ~6 months part). Two samples using deuterium labelled Bcl-2 in a h-DMPC matrix (Fig 2 and Fig S1) and one using h-Bcl-2 in a deuterium labeled DMPC matrix (Fig S2).

It should also be noted that the DMPC only example in Figure 2 is not a DMPC membrane prior to the interaction/penetration of Bcl-2, but rather a protein free membrane shown to highlight how the presence of Bcl-2 within a membrane composed of the same lipid modifies the experimental reflectivity data sets and resulting scattering length density profiles.

2. There is a discussion about the conformational rearrangement required for a protein to convert from a water soluble form to a membrane embedded form but this is too brief to explain to the reader the author's conclusions as to the crucial event which is being characterised here.

Our reply: In the introduction (lines 58-95 in revMS) we have now clarified the general behavior of "tail-anchored" membrane proteins of the Bcl-2 family and their posttranslational transport from the ribosome via the cytosol to the target membrane under conformational rearrangements. While (as mentioned now in more detail), various of those Bcl-2 proteins like Bcl-x_L, Bax etc. are rendered soluble by hiding their C-terminal hydrophobic anchor in their BH3 binding groove prior membrane insertion, full length Bcl-2 protein being studied here is not found in the cytosol (s. new Ref. 16 in revMS) since it presumably does not hide its TM region. Therefore its in vivo transport to its target membranes is not clear at all and various proteins like mitochondrial chaperones proteins are discussed in this context (s. ref. 20, and 16 in revMS). We also generated a Bcl-2 Δ TM construct which we could show was partially soluble (revised Fig SI6) and underwent typical conformational changes in the presence of membrane-mimicking detergent DPC (in depth discussion lines 382-414 in revMS); changes (s. Fig. S6) similar as observed for Bcl-x_L and Bcl-w proteins (s. ref. 45,46 in revMS) under those conditions. The Bcl-2 Δ TM construct was fully functional under those conditions like the intact Bcl-2 by testing Bax-BH3 binding (s. Fig. SI 9 in comparison with Fig. SI3 in revSupp). As also seen in SI7, SI8 the general structure of this construct is also in good agreement with the general structure obtained for intact Bcl-2 in those environments.

3. Firstly, the language needs to be tightened up for example Line 308 "Generally, the N-terminal protein heads of these Bcl-2 members are amphitropic (soluble or membrane-bound)". Amphitropic describes the affinity for both water and membranes not the state of being bound, and what is a N-terminal protein head?

Our reply: We have now corrected those inaccuracies and used also either N-terminal head domain/common globular fold throughout the text.

4. The critical event here is that the amphitropic helices will need to rotate to expose their hydrophobic sides to the membrane rather than the protein core and the hydrophilic sides will now form a hydrophilic core. This might be expected to increase the water content of the membrane unless a highly defined new set of hydrogen bonding interactions takes place to satisfy the polar interactions of all the hydrophilic side chains. In most previous cases of proteins inserting into the membrane like this the resulting confirmation of the altered membrane embedded form In fact the membrane embedded form of BCL-2 has even introduced channels in the lipid bilayer PNAS May 13, 1997 94 (10) 5113-5118, Biophys J Volume: 104 Issue: 2 Pages: 421-431 (although not as clear as the oligomeric Bax).

The structure being presented here is therefore a novel form of membrane embedded protein and should therefore be discussed in clearer and more precise detail as the authors explanation of this truly revolutionary structure.

Our reply: We thank the referee for highlighting those publications which are now together with a more in depth comparison of the membrane-embedded structure of our full-length Bcl-2 and the truncated soluble constructs that can form channels included in revMS (lines 352-364) and ref. 43,44 in revMS). As mentioned under 2. We included also references which suggest involvement of other proteins (s. ref. 16,20 in revMS) in Bcl-2 translocation to the MOM.

With respect to the water content please see reply above to point 1.

5. Minor points figure 2

Figure 2 What is the 4th solution condition in figure 2b with Bcl2 also please explain the flat base to the data fit in the pure DMPC data.

Our reply: We thank the reviewer for noticing that that label is missing for the 80% D2O data set in the SLD profile in Fig. 2B. This has now been corrected.

The flat base in the DMPC only data shown in Fig 2B is due to the SLD of the tails matching very closely to water SLD.

6. The description of the data fitting in Results & Materials and methods is confusing . In the first “The analysis was carried out using Abelès matrix formalism, which divides the focal (why focal?) membrane into discrete homogeneous layers across the bulk interface”. In the second “NR data were analyzed using RasCal software..., which fits layer models describing the interfacial structure calculated using Parratt’s recursive formalism to experimental data. In this approach, the interface is described as a series of slabs, each characterized by its nSLD, thickness and roughness”. Is this the same approach?

Our reply: We thank the reviewer for highlighting this mistake. RasCal using the Abelès matrix calculation. Though conceptually similar, Parratt’s recursive formalism (old ref 50 now removed) and Abelès matrix calculation are mathematically distinct. This fact has now been corrected in the manuscript (line 626 and ref 33 in revMS). In addition, we removed the somehow misleading “focal” expression (see lines 125 and 181 in revMS).

REVIEWERS' COMMENTS:

Reviewer #2 (Remarks to the Author):

The authors have replied fully to my questions and the changes to the manuscript satisfactorily address all the issues raised

REVIEWERS' COMMENTS:

Reviewer #2 (Remarks to the Author):

The authors have replied fully to my questions and the changes to the manuscript satisfactorily address all the issues raised.

Reply: We are happy with the reviewer's final evaluation of our revised work.